# Benefits of dance for Parkinson's: The music, the moves, and the company

Corinne Jola[1]*, Moa Sundström[1], Julia McLeod[2]

**1** Division of Psychology and Forensic Science, School of Applied Sciences, Abertay University, Dundee, United Kingdom, **2** Division of Nursing and Mental Health, School of Applied Sciences, Abertay University, Dundee, United Kingdom

* c.jola@abertay.ac.uk

**Data Availability Statement:** Not all data can be shared publicly because of sensitive information in the qualitative data set. The core data and analysis process in R underlying the quantitative findings presented in the study are available from the

## Abstract

Dance classes designed for people with Parkinson's are very popular and associated not only with increasing individuals' motor control abilities but also their mood; not least by providing a social network and the enjoyment of the music. However, quantitative evidence of the benefits is inconsistent and often lacks in power. For a better understanding of the contradictory findings between participants' felt experiences and existing quantitative findings in response to dance classes, we employed a mixed method approach that focussed on the effects of music. Participant experience of the dance class was explored by means of semi-structured interviews and gait changes were measured in a within-subjects design through the Timed Up and Go (TUG) test before and after class, with and without music. We chose the TUG test for its ecological validity, as it is a simple test that resembles movements done in class. We hypothesised that the music and the dance class would have a facilitating effect on the TUG performance. In line with existing research, we found that before class, the gait of 26 participants was significantly improved when accompanied by a soundtrack. However, after class, music did not have a significantly facilitating effect, yet gait without music significantly improved after class compared to before. We suggest that whilst the music acts as an external stimulator for movement before the dance class, after the dance class, participants have an internalised music or rhythm that supports their motor control. Thus, externally played music is of less relevance. The importance of music was further emphasised in the qualitative data alongside social themes. A better understanding of how music and dance affects Parkinson's symptoms and what aspects make individuals 'feel better' will help in the design of future interventions.

## Introduction

Parkinson's is a progressive neurodegenerative disorder that affects more than 10 million people worldwide [1]. The disease manifests itself predominantly via a range of motor symptoms, such as tremor, slowness of movement (i.e., bradykinesia), freezing, painful muscle cramps (i.e., dystonia), and stiffness [2] but also other autonomic and sensory nervous system symptoms manifest themselves, such as problems with sleeping [3] or dizziness [4]. Parkinson's is

https://osf.io database (DOI 10.17605/OSF.IO/Q5GE8).

**Funding:** Part of the project was funded by a Carnegie Vacation Scholarship scheme (Nr VAC008809) awarded to MS with CJ as their supervisor and by the Scottish Funding Council (Nr 2756) to boost research and knowledge exchange in response to the disruption by the COVID-19 pandemic awarded to CJ via Abertay University. The funders had no role in study design, data collection and analysis, decision to publish, or preparation of the manuscript.

**Competing interests:** The authors have declared that no competing interests exist.

characterised by a reduction of dopaminergic cells in the substantia nigra, located in the mid-brainstem and responsible for functions such as motor control, sleep-wake rhythm and excitement. The depletion in dopamine obstructs the control of voluntary motor processes, in particular the deliberate initiation of a movement. This is because the signal to execute an action is controlled by the basal ganglia, a structure whose functions are interlinked with the substantia nigra. However, the visible clinical symptoms differ from patient to patient and today Parkinson's is recognised as a heterogenous condition that includes physical as well as psychological symptoms. The latter show themselves in form of depression, anxiety, cognitive impairment, and more, together leading to a generally reduced quality of life [5, 6]. Despite advances in the understanding of the underlying mechanisms of Parkinson's, including the identification of potential genetic mutations [7], there is currently no cure and the heterogeneity of the symptomatic is challenging research studies and interpretations of the findings.

At present, the most common treatments are medical dopamine replacement and the application of deep brain stimulation [8]. The former is problematic since the effectiveness of the medication declines over time and people report unwanted side effects. However, whilst brain stimulation reports some success, it requires further research regarding the efficacy of the method in relation to alleviating negative side effects associated with deep brain stimulation [9]. Notably, there are several alternative treatments which help managing Parkinson's symptoms, such as exercise, speech and language therapy, as well as cognitive training [10]. Likewise, dance, music, and singing therapies [11] have received much attention for their consistently perceived positive impact on individuals' quality of life. Here, we focus on the effects of dance classes for which empirical evidence on motor as well as non-motor control improvements are however mixed [12, 13]. It is thus of importance to better understand some of the mechanisms behind the changes observed through such interventions. Since dance classes combine a multitude of elements, it is important to look at these individually as well as consider the experience of people with Parkinson's engaging in those activities when designing studies and interpreting the findings. The integration of movement and music are most prevalent and will be explored here more specifically–yet other social and emotional aspects are also crucial in dance and its impacts on people with Parkinson's were previously reported [14].

Music-based movement therapies have repeatedly been found to improve both cognitive functioning as well as life satisfaction in those with Parkinson's [11, 15]. Music and singing are thought to be beneficial as they can be used as a tool to communicate and express emotions, both of which can be difficult for people with Parkinson's [16]. This is especially important, as the symptom of facial masking in Parkinson's (often leading to an inability to express emotions) can have a negative impact on close relationships [17]. These kinds of therapies could therefore be a way of helping individuals with Parkinson's to maintain healthy relationships, keep an active social life, and therefore maintain their quality of life.

On a neurobiological level, the benefits of music are twofold. Firstly, the general positive feelings associated with music [e.g., 14] facilitate dopamine release through increased activation of the limbic system in the general population [18] and could thus be effective in people with Parkinson's also. Notably, however, whilst people with Parkinson's commonly report a calming effect, they do not seem to subjectively experience an improvement in their motor symptoms after listening to music alone [19]. Secondly, there is evidence that a musical cue can serve as an external stimulus to initiate movement in Parkinson's sufferers [20]. Thus, it may be the combination of music and movement that is of relevance as subjective improvements after dancing, for example, are vast [21], and empirical evidence of rhythmically cued movements supports this. For instance, Ghai and co-authors [22] showed that external auditory cueing consistently increased participant's speed and stride length. It is suggested that by partially overlapping neuronal activation of internally controlled movement and externally

evoked actions, it seems possible to bypass the function of basal ganglia with external stimulation [23, 24]. Such evidence proposes that Parkinson's sufferers may gain better control of movement when they move to music with a clear rhythmical beat. Accordingly, Nieuwboer et al. [25] found significant improvements in gait and balance in a single-blind randomised crossover trial intervention based on a home rhythmical cueing programme. Notably though, whilst the impact of rhythmical cueing decreased with time, gains of a tango dance class intervention were maintained in the one-moth follow -up [26].

It is important to remember that the alternative therapy of dancing combines the physical, rhythmical, psychological and emotional aspects of the above alternative treatments [27]. Hence, it comes as no surprise that dance classes were found to help manage Parkinson's symptoms and improve the individuals' quality of life as well as their motor control which could all contribute to the prolonged benefits. Rocha and co-authors [28] studied mobility effects of the Argentine Tango and mixed dance in Parkinson's and found that while Tango improves a person's general mobility and balance, the mixed dance programme decreased freezing of gait after dancing for eight weeks.

Evidence from qualitative studies has indicated that the majority of participants experience these types of community-based exercise dance programmes as helpful in relation to regaining a sense of positive identity after a diagnosis of Parkinson's, decreased feelings of isolation, better mood and quality of life, reduced motor symptoms, and enhanced functioning [29–32]. These gains have been reported to be experienced by participants as persisting at 6-month follow-up [33]. Although the opportunity for social contact is highlighted by many dance class members as being a major factor contributing to improved mood and well-being, similar positive experiences have been described by participants in an on-line dance class characterised by minimal opportunity for social interaction [34]. A common theme in all qualitative studies of the lived experience of people with Parkinson's disease is that they are keen observers of their own functioning, appreciate comparing their disease stage with others through observation, and are highly motivated to share their experiences with researchers. As a result, the inclusion of qualitative methodologies in research on the effects of dance interventions on functioning and well-being in patients with this disorder has the potential to identify change processes and outcomes that may be hard to observe using standardised quantitative measures alone. We have therefore included a qualitative element to the study. Namely, despite the positive testimony of patients and carers who have participated in dance programmes, evidence from quantitative empirical studies presents a mixed picture. Numerous studies on the effects of dance for people who suffer from Parkinson's have been conducted and whilst many found psychological benefits [35–37] and evidenced improvement on motor and cognitive symptoms in their meta-analyses [38, 39], the results are not conclusive or powerful enough to inform practice (see for example [40]). Strikingly, Earhart and co-authors did not find any significant advantage of the dance practice over another physical activity (i.e., treadmill) in their large-scale study behavioural and fMRI study [41, 42]. In fact, only the Parkinson's participants of the control intervention (treadmill and stretching) showed significant improvements in forward and backward walking, whereas the Parkinson's participants of the dance intervention group practicing tango, did not. Moreover, no significant changes in cortical activity during motor imagery in response to exercise were identified. One explanation for the mixed findings in changes associated with Parkinson's dance classes is that certain aspects of dance might not have been considered in the assessment, thus affecting some measurements. For instance, since most dance classes for people with Parkinson's are held in groups, individuals also have the opportunity to practice motor observation. People with Parkinson's do report enjoying action observation in dance classes [14]. Moreover, action observation is a recently developed form of treatment for Parkinson's. In therapies targeting action observation, even simple

actions such as rising from a chair, or slow walking, appear to evidence benefits in many typically impaired domains [43]. These include sustained improvements to gait, walking speed [44], turning, and balance, [45], lasting up to 3 months. Since action observation is an intrinsic element of dance classes, it is not clear to which extent it contributes to the benefits of dance. Moreover, Fontanesi and DeSouza [46] suggested that dance as an exercise is unique for its intrinsic artistic elements that may influence affective, aesthetic, and motor responses. As the qualitative reports are consistently highlighting positive impacts of dance, it is important to better understand the role of dance practice for Parkinson's. What is evident from the existing studies is that a) many employ a different dance style, b) follow different intervention duration and c) employ different measurement types and analyses. To advance our understanding of the impact dance classes have on people with Parkinson's, we thus conducted a mixed-method study that made use of both qualitative and quantitative data to ensure that we capture both quantified behavioural responses and the experience of dance. This is motivated by the recognition that the experience of a dance class and behavioural responses are intertwined. An interdisciplinary approach combining qualitative and quantitative observations can thus provide further insight into the mechanisms supporting benefits of dance. We consider this approach particularly relevant in studies with people with Parkinson's, since the disease is heterogeneous. As the focus of the study's design was high ecological validity, data collection related to performance was thus kept as close as possible to the activity people with Parkinson's engaged in during the dance classes and with that, the potential contributing factor of music and a simple TUG test took centre stage. The qualitative data were collected with the aim to gain an indepth understanding of the experience people with Parkinson's have in the dance classes, helping with the interpretation of the experimental findings but also assessing whether ecological validity was met.

The current study thus explored whether music is a contributing factor of Parkinson's dance classes that benefit individuals with an immediate change in their motor ability after dancing. At the time of our testing, to our knowledge, our study was novel in that we conducted a within-subjects repeated measures mixed-method design in a Parkinson's group on the effects of dance, combining qualitative and quantitative outcomes of participants that would normally partake in these classes.

## Data collection

### Dance class

Our participants were recruited from overall six different locations each part of established dance programmes for people with Parkinson's (i.e., South London Inclusive Dance Experience, Move into Wellbeing, Dance for Health, and Dance for Parkinson's Scotland). Whilst the dance classes might have slight variation and different teachers, they shared that the accompanying music was predominantly recorded pop and rock music played through loudspeakers. In all classes, the movements were adjusted to the participants needs and abilities and included arm and feet movements while sitting, clapping and facial expressions and movement from the back of the chair (with support) and around the space when possible. One location engaged participants in dancing as well as singing, however, these participants were not included in the quantitative study. Further, whilst some teachers have trained in Dance for PD, at the time of the study, only one of these classes were an international associate hub from the Dance for PD classes, a programme founded by D. Leventhal and his team at Mark Morris dance. All participants participated in the dance classes at least once a week with on average of a total of over 40 dance classes. Only four participants danced for less than two months. Three of those were included in the quantitative study.

Table 1. Participant demographics.

| Descriptive Variables | Subjective Experience | | Timed Performance | |
|---|---|---|---|---|
| | Males (n = 6) | Females (n = 5) | Males (n = 11) | Females (n = 15) |
| Age (years) | 74.5 ± 8.4 | 68.8 ± 11.9 | 70.7 ± 8.4 | 72.1 ± 8.7 |
| Nr of Dance Classes | 62.0 ± 56.2 | 70.2 ± 128.6 | 69.4 ± 70.3 | 58.4 ± 83.3 |
| Time since PD diagnosis | 4 ± 2.53 | 10.3 ± 8.1 | 7.8 ± 7.6 | 6.9 ± 5.9 |
| Hoehn & Yahr scale | 2.25 (1–4) | 3 (1.5–4) | 3 (2–4) | 1.5 (1–4) |

Mean ± SD for Subjective Experience (Qualitative approach) and Timed Performance cohorts (Quantitative approach, Timed Up and Go test; TUG); except for Hoehn & Yahr scale (Median and range [47]). The Hoehn & Yahr scale has 2 missing data points (1 male, 1 female in Study 2). Nr of dance classes are an estimate based on the reported number of months of dance classes attended and average number of weekly classes taken. Time since PD diagnosis in years.

## Qualitative approach

**Participants.** Six female and 5 male participants were interviewed, aged from 50 to 84 years, reflecting a range of severity, chronicity, pattern of illness presentation and work status (see Table 1). Several participants reported involvement in other well-being activities, such as yoga, acupuncture, exercise classes, Pilates and counselling. 8 out of the 11 participants interviewed participated also in the quantitative study. Participants who volunteered to be interviewed were based in the North of Scotland and London. All participants provided written informed consent as approved by our local University ethics board (EMS1636).

**Procedure and data analysis.** A semi-structured interview guide was used to facilitate an open exploration of the lived experience of dance class participants. Individual interviews were carried out after the dance class, in a quiet space at the dance location. Interview questions invited research informants to talk about their living situation and the effect that Parkinson's has had on their lives, and their involvement in the dance class. Interviews were audio-recorded, transcribed verbatim by the second author, and subjected to thematic analysis in accordance with the procedures specified by Braun and Clarke [48]. The qualitative data was gathered and transcribed by the second author and analysed by the last author, who was blind to the hypothesis and aims of the study (i.e., had no knowledge of the quantitative design or the questions regarding the music vs. no-music effect of dance on Parkinson upon identifying the themes). A theme is defined as a construct that "captures something important about the data in relation to the research question and represents some level of patterned response or meaning within the data set" [48, p. 82]. The analysis of the transcript data was according to the following steps: reading through all the transcripts to become familiarised with the data; coding statements in terms of units of meaning that represented different aspects of participant lived experience; reviewing codes to identify themes/patterns; defining and naming themes.

## Quantitative approach

**Participants.** 15 female and 11 male participants between 50 and 84 years of age (M = 71.31; SD = 8.39) completed the Timed Up and Go test (TUG) (see Table 1). All but one participant were on a settled timed medication schedule. All participants met the inclusion criteria in the TUG analysis of (a) a diagnosed PD and (b) an average TUG time before the dance class (without music) > average TUG time + SD for the age-matched general population [49, 50]. Due to resource limitations, it was not practical to set a sample size based on a priori power analysis. Our sample was therefore a convenience sample. We aimed for a cohort that was equal or larger in size than the existing literature with similar designs [51]. We also ensured that all the assumptions were met for our analyses. All participants provided written informed consent as approved by our local University ethics board (EMS1636).

**Procedure and experimental design.** The experiment consisted of 2 conditions, i.e., music (perform TUG with and without music) and time (before and after class). Participants completed three trials per condition. The order of the conditions was counterbalanced across participants, leading to 4 order types.

*Music.* The music used in the music condition was Dancing Queen by Abba, which is described as a Europop version of American disco music [52]. The song is popular worldwide and one of the most recognised ABBA songs. Participants instantly recognised the song which was played around 35 seconds into the track. It is considered as energetic [53] and emphasises the theme of the joy of dancing in its lyrics. It has 100 BPM, which resembles a brisk walking pace [54]. It is thus one of the songs that are often played during dance for Parkinson's classes.

*TUG.* The TUG was administered as an indicator of daily living ability [55]. The instruction to participants was to rise from a chair (without using their arms) on a verbal cue, walk three meters (indicated with a coloured tape on the floor), turn around and walk back to the chair. Participants were advised to do it at a comfortable pace as they would normally walk to elicit natural responses and prevent falls. The time was taken from the moment participants lifted their weight from the chair to the touch back and it was explained to participants that we measure time as the speed is important.

**Data analysis.** The effects of music (music vs. no-music) and dance (before vs. after) on TUG times were analysed by means of the linear mixed effects model lme4 package in R (version 4.12 [56]). We first examined whether the data met the normality assumption, by means of visual inspection of the residual scatter plots (S2 Fig) as well as formal testing through Shapiro-Wilk tests of the residuals. We then calculated Cook's distances of the residuals to classify possible influential outliers. We chose to investigate our data for outliers since our cohort were elderly participants who might have misunderstood the task or were otherwise influenced by distractors in certain trials. However, our participants are also diagnosed with Parkinson's and therefore variances within and between participants were expected and the general 4/n cut-off point for Cook's $D_i$ too sensitive. We therefore chose a more conservative Cook's cut-off point of $> 4^*$mean [57] that is still more sensitive than the otherwise suggested $>1$ in lieu of 4/n [e.g., 58]. After outlier removal, we re-fitted the linear mixed model using the restricted maximum likelihood method (REML) on the loglinear transformed TUG times, based on the null-model without the fixed effects of music and time (1 + (1|p)). When normality of the underlying residuals could be assumed, we compared the null-model with the alternative 'full-model' including the predictors conditions music and time as fixed effects and participants as a random nested effect (M*T + (1|p)). The better model was selected based on Akaike's Information Criterion (AIC [59]). When an interaction of that model was noted, Tukey post hoc tests were used to determine group changes from baseline. Statistical significance was set a priori at $P \leq .05$. Effect size measures (ES) were used to show changes in terms of standard deviation.

**Hypotheses.** We tested three predictions. First, that music facilitated performance of the TUG as would be indicated by a main effect of music (i.e., less time with than without music, H1). Second, that the dance practice had a facilitating effect, which would be evidenced by a main effect of time (shorter TUG after vs. before class, H2). The third hypothesis addressed the ecological validity of the testing design, which was that through dance practice, participants improve their performance in the way they have been dancing, i.e., with music but not without. The latter would be evidenced by an interaction (H3), with decreased time after the dance class in the presence of music but no change or increased time after the dance class without music.

# Findings

## Subjective experience

Qualitative analysis of interview transcripts generated two broad themes as illustrated in Table 2: the effect of taking part in the dance class, and helpful aspects of the dance class.

**The perceived effects of taking part in the dance class.** All interviewees reported that their participation in the dance class had been a positive, valued experience. The only reports of a negative effect were from one person whose pre-existing back pain was sometimes exacerbated by the class. Sub-themes within participant accounts of the effects of dance included: social contact, exercise, positive well-being, and functioning. Informants believed that the benefits of taking part in the dance class consisted of both short-term and longer-term gains.

*Social contact.* All informants highlighted the extent to which the dance class provided them with an enjoyable social experience. For a few, with limited social contact in their life, the class represented a weekly event to which they looked forward with pleasure. Fellow class members were described as 'friends' and 'nice people'. Some informants used the term 'comradeship' in relation to the social aspect of the class, to convey a sense of being on the same side in a battle against a disease. A few talked about how people in general did not understand the complexity of Parkinson's, and that it was a relief to be able to spend time with others who appreciated what they were going through. Social contact did not just refer to the experience of participating together in dance, but also to the possibility of meeting new people and talking with fellow members before and after sessions.

*Exercise.* All informants value the opportunity to engage in vigorous exercise on a regular basis. Some described themselves as 'feeling looser' and more relaxed, being able to move and breathe more easily after each class and "tired in a nice way".

*Positive well-being.* Attendance at the dance class was reported as a pleasurable experience that helped participants to feel good about themselves. Informant described this as a sense of being 'uplifted' or 'high'. Others talked about 'endorphins' being released. Another commented that the class was a source of enjoyment, whereas doing exercise at home was generally boring. A few participants mentioned that the dance class had helped them to combat periods of low mood: "it cheers you up a bit", "helps you to get on with the day", "makes me feel worthwhile", "you feel better".

*Functioning.* Some participants described specific ways in which involvement in the dance class had improved their motor functioning and co-ordination. One reported that "I often go home and cook a dinner rather than get something out of the freezer. . .[Having Parkinson's means that] its very difficult peeling things and so you might as well do a lot while you are at it". Another talked about using a mental image, through dancing, of a relaxed motion sequence

**Table 2. Qualitative main and sub-themes.**

| Main Themes | Sub-themes |
|---|---|
| The effect of taking part in the dance class | • Social contact |
| | • Exercise |
| | • Positive well-being |
| | • Functioning |
| Helpful aspects of the dance class | • Music and rhythm |
| | • Supportive and encouraging teachers |
| | • Movements that exercise all parts of the body |
| | • Explanations of the purpose of different routines |
| | • Getting in touch with previous treasured life experiences |

of putting his foot down in a certain way, that enabled him to walk at times when he would otherwise have been unable to do so. This informant added that "after just two minutes [dancing] I tend to walk pretty normal—dancing and music are definitely the key to Parkinson's illness".

**Helpful aspects of the dance class.** The majority of participants were able to identify specific ways in which routines and activities in the dance class had been helpful for them. The main sub-themes in this domain were: music and rhythm; supportive and encouraging teachers; movements that exercise all parts of the body; explanations of the purpose of different routines; getting in touch with previous treasured life experiences.

*Music and rhythm.* The fact that the dance class offered exercise and movement that involved music was viewed as particularly important by several participants. One reported that "having the music and the rhythm helps me to take things calmly and not rush" and that the music had a "liberating" effect on their capacity to move. Another remarked that "when you get the music in your head you can move". The presence of live music in some classes was regarded as particularly helpful.

*Supportive and encouraging teachers.* All participants referred to the encouraging, sensitive and responsive teaching style of the dance class facilitators, and the ways in which they used humour to make the class an enjoyable experience: "we do exercises that are really very witty".

*Movements that exercise all parts of the body.* Several participants reflected that it was particularly helpful to have an exercise routine that required them to move all parts of their body: "it works all parts of the body and I think that makes a lot of difference because when you've finished you feel as if you've moved every joint–not unnecessarily but in moving every joint it makes you feel a little bit easier. . .every muscle has been tweaked".

*Explanations of the purpose of different routines.* An aspect of the class that was described as helpful by some participants was the way that the teachers explained the purpose of each routine that they introduced: "they stop at each one and explain the exercises and it gives you a chance to understand what the exercises are for . . ..they don't just do it for the sake of it, it's there for a reason". Participants reported that this information helped them to make use of exercises in everyday situations.

*Getting in touch with previous treasured life experiences.* A few participants talked about how the dance class helped them to recall and re-live periods in their life where they had either been a dancer, or had engaged in a similar activity such as marching. These connections gave the class added meaning.

## Timed performance

A total of 15 Cook's $D_i$ from the null-model's residual were above 4 times of the mean (>0.358) and those observations were thus removed from the analysis (see S2 Fig), corresponding to 4.81% of the total observations (312). The outliers were spread across 9 participants and only in one case two trials were removed from the same condition (before dance, with music). All but two of these observations were also beyond the cut-off point in the full model.

The repeated analysis without those outliners of the null-model's linear mixed model fit by REML on the loglinear transformed TUG time and participants as a random nested predictor showed that normality could be assumed (Shapiro-Wilk = 0.997, $p$ = 0.879, see also S2 and S3 Figs). The null-model (Residual = 0.0835; REML criterion at convergence = -505.96) showed a significant fixed effect, $t(25.00)$ = 47.27, $p <$. 2e-16.

The alternative full model including Music and Time as fixed effects (Residual = 0.0067; REML criterion at convergence = -500.5) showed a significant effect of music and time, $t$

**Table 3. Descriptive data.**

| Music | Time | Mean | SE | Confidence Interval 95% | |
|---|---|---|---|---|---|
| | | | | Lower | Upper |
| Without Music | Before dance | 2.48 | 0.0526 | 2.37 | 2.59 |
| With Music | Before dance | 2.45 | 0.0526 | 2.34 | 2.55 |
| Without Music | After dance | 2.44 | 0.0525 | 2.33 | 2.54 |
| With Music | After dance | 2.43 | 0.0525 | 2.33 | 2.54 |

Descriptive statistics using estimated marginal means for the linear mixed model with music and time as fixed effects: Reported are mean and standard error (SE) of the log-transformed TUG times for all fixed effects individually according to Kenward-Roger methodand 95% Confidence interval.

(268.03) = -2.60, $p < .0099$ and $t(268.03) = -3.36$, $p < .0009$, respectively. The interaction in the model did not reach significance, $t(268.02) = 1.76$, $p = .0796$. AIC analysis shows that the full model (-513.66) predicts the data significantly better than the null-model (-504.06), Chisq (3) = 15.58, $p < 0.0014$. REML of the full model showed that the TUG with music models a significant increase in performance (i.e., reduces TUG time) and so does the dance class intervention (Table 3, see also S1 Table for descriptive statistics on the original data). However, contrast of the main effects showed that music was borderline significant and further exploration of the pairwise effects showed that time influenced performance without music but not with (Table 4).

## Discussion

Our mixed-method study on the impact of music before, during, and after dance for Parkinson's classes partly supported our hypotheses and are in line with existing qualitative observations. In our study, we measured people with Parkinson's performance in the Timed Up and Go (TUG) test as an indicator of their everyday life activity levels and motor control and enquired their individual experiences of the dance classes and its benefits.

### Qualitative analysis of participant lived experience

Our qualitative data add to our understanding of the importance of music as an element of the dance class experience as a whole. While our participants emphasised that music and rhythm were helpful aspects of the dance classes, other elements were equally important, such as social contact. Indeed, music and movement together make a community: Dunbar [60] proposed that dance and music developed in the history of human evolution because the social groups

**Table 4. Contrast main and pairwise effects.**

| Contrast | Estimate | SE | p |
|---|---|---|---|
| Music (Without music–with music) | 0.019 | 0.0095 | 0.0506 |
| Time (Before Dance–after dance) | 0.029 | 0.0095 | 0.0029 |
| **Pairwise effect** | **Estimate** | **SE** | **p** |
| Before Dance (Without music–With music) | 0.03531 | 0.0136 | 0.0482 |
| After Dance (Without music–With music) | 0.00194 | 0.0132 | 0.9989 |
| Without Music (Before dance–After dance) | 0.04519 | 0.0134 | 0.0049 |
| With Music (Before dance–After dance) | 0.01182 | 0.0134 | 0.8144 |

SE (Standard Error) and significance level for each main effect (df = 268 for all contrasts) obtained through least-square means contrasts for the linear mixed model with music and time as fixed effects.

became too large for one-to-one grooming and a more efficient way to keep the social structure was necessary. Hence, according to this theory, the effects of music and movement are not dissociable from the effects of feeling part of a group. Hence, it is no surprise that our participants mentioned all of these. Nevertheless, participants were able to identify a wide range of specific helpful aspects of their involvement in the dance class. Several participants mentioned that the enthusiasm and supportiveness of the dance teachers, and their ability to provide clear explanations of the rationale for different activities, were particularly valuable–a theme that has also been highlighted in previous studies [14, 28]. Moreover, considering the importance of the group and our aim to follow an 'ecological approach' one might want to consider providing participants with the opportunity to be tested in a group setting. Notably, several studies emphasised the importance of being able to observe others [14], so much so, that action observation has been developed as a therapeutic tool for people with Parkinson's [61]. Yet surprisingly, observing others' moves were not part of these individuals' reported experience.

## Quantitative results of model predictions on performance

Regarding the experimental part of our study, our first hypothesis was that music facilitates TUG performance. Accordingly, we found a significant main effect of music in our model. The facilitating effect of music was also confirmed for the contrast analysis before the dance class (music vs. no music), however, the beneficial effect of music compared to no music after the dance class was not evident. Numerous experiential and qualitative studies have found that music can facilitate motor initiation in people with Parkinson's [e.g. 11, 62, 63]. Yet, Bella and co-authors [64] showed that cued gait performance was not significantly improved in people with Parkinson's after gait training with rhythmic auditory stimulation whereas un-cued performance was. However, to our knowledge, no study has yet investigated the impact a dance class intervention with music has on this well-known facilitation of music in thought of the conflicting after-effect identified by Bella et al. [64].

Our second hypothesis predicted that the dance classes improved motor control for people with Parkinson's. This was confirmed by the significant main effect of time in our model. Moreover, TUG performance was significantly improved through a dance class as found in our contrast analysis. Notably, a dance class with music significantly improves motor control in people with Parkinson's without music up to a similar level as when using music without dance interventions.

Our model did not render the interaction of the fixed factors music and time significant. However, as discussed above, the contrast for music alone was borderline significant and subsequently conducted pairwise contrasts showed that the direction of the effects was contrary to our predictions. We hypothesised that testing the TUG in an ecologically valid manner (i.e., with music, to align with the dance classes) will further increase TUG performance after the class compared to a less ecologically valid manner (i.e., without music). Our TUG measures showed that it is unlikely that the TUG performance with music after dancing was beneficial and that there is strong evidence that performance after the dance class without music was at the level of performance with music at any time point similar to the observations by Bella et al. [64]. This finding can thus be explained in two possible ways: Firstly, by a 'ceiling effect' in that participants might have reached a performance threshold when moving with music that cannot be improved further. Secondly, participants may have internalised music from the dance class and therefore did not require additional external stimulation to perform better, as they would before the class. For the former (i.e., threshold explanation), an additional model including the experience of the participants in form of the estimated hours of dance classes was explored (S1 File). The observations indicate that participants with more dance experience

show stronger evidence for long-term than short-term effects. This is in line with the previously reported higher sensorimotor synchronisation ability through dance experience in Parkinson's [65]. However, this exploratory model must be considered with caution. The amount of expertise (hours participated in dance classes) varied widely and were based on participants' estimated values. These are likely influenced by subjective perceptions and a more controlled approach is required for verification.

To conclude, it is likely that through participating in dance classes, people with Parkinson's can internalise music in their minds [66] which help them further increase the benefits of the class. Several qualitative studies suggest that people with Parkinson's internalise music to their advantage in everyday life [e.g., 14, 67]. However, when then presented with an external sound on top of the internalised beat, this could possibly lead to cognitive interference. According to Lavie [68] auditory distractions can disrupt task performance if the exercise mode and/or intensity demand high levels of concentration as is the case for people with Parkinson's. Whereas under normal circumstances, parallel processing allows individuals to ignore external stimuli [69]. Hence, while people with Parkinson's use the music before class as an external stimulus, after class, participants initiated the movements internally, and similar to a difficult task for healthy participants, the music acts less as an auditory distraction the more dance experienced individuals are. We did however not find evidence that the internalisation was a source of auditory distraction, as showed by Bigliassi et al. [69].

Music-related interventions are common in the field of exercise and sport enhancement; assumed to enhance performance by reducing dissociative thinking [70, 71]. For people with Parkinson's, an external rhythm is employed as an indirect stimulation. However, the durational effect of this is currently not understood. Contrary to our interpretation of internalising the music, it is generally understood that stopping the external cueing immediately reduces the effects, which suggests that constant cueing is necessary [72]. However, the authors employed metronome embedded music which might have been too complex a stimulus to be internalised. Moreover, music is known to be a better stimulus for entrainment than a metronome [73] and in the study by Nieuwboer et al. 2007 [25], the effects of a cued home exercise declined over an extended period of 6 weeks. In addition, Harrison and co-authors recently showed that participants with Parkinson's who respond positively to self-generated rhythmic auditory cues are able to induce similar increases in motor movement through these as can otherwise be achieved by externally-generated cues [74].

Yet, the use of music is hypothesised to have a detrimental effect on motor performance if the exercise mode demands high levels of concentration. In such instances, the human brain attempts to partially suppress or parallel process potential distractors to enable the organism to engage with the task. The underlying mechanisms of parallel processing have not been investigated extensively during the execution of motor tasks [e.g., 75]. It is unclear as to whether the music we played in the testing phase might have impacted on a further increase in performance due to mismatching rhythm. Ashoori et al. [76] has postulated that an adaptive system that adjusts the rhythm of movement is most efficient for intervention in people with Parkinson's. In fact, in Parkinson dance classes, emphasis is placed on accompanying the dancers with live music, allowing for immediate adaptation of rhythm and melody, and optimising the movement performance. It is suspected—but has barely been studied [64]—that these effects are inconsistent, as people with different capacities presumably have individual rhythm requirements [22].

## Limitations of our study

It is important to interpret the qualitative findings with caution. While research informants engaged openly and honestly with the interview process, interviews were relatively brief–it is

likely that participants would have been able to generate more detailed accounts of their experience if they had been able to engage for longer, at different points over the course of the programme, or possibly if interview questions had been provided in advance. The interview schedule was intentionally open-ended, as a means of encouraging informants to express their own personal points of view and reflections. However, it could have been useful to have added probe questions that asked about specific aspects of the dance experience, such as the choice of music. Despite being specifically asked about less satisfactory aspects of the dance class, participants did not report any negative experiences. Interviews with individuals who had dropped out of the class might have enabled this topic to have been explored more fully. It would also have enhanced the credibility of the qualitative analysis if it had been possible to invite participants to comment on findings. Follow-up interviews would have made it possible to explore the extent to which participants experienced sustained benefit, and the strategies they used (e.g., self-initiated dance at home) to maintain these gains.

Another limitation of our study is that we had resources limitations that restricted the number of participants. It can thus be argued that our design is underpowered–or in other words, less sensitive to our alternative hypotheses H1, H2, and H3. The risk for falsely rejecting small to medium effects is indeed relatively high with a small sample size. The results of our analyses should thus be interpreted with caution when even small effects are effects of interest as they are here; and this is why we conducted post-hoc contrast analyses despite the interaction of music and time not reaching significance in the mixed linear effects model. Clearly, our study would benefit from replication in form of a clinical trial.

Also, one could argue that our design misses a control group. Whilst we agree that a control group increases internal validity, external validity, a core focus of our study, is reduced. Another issue is the selection of the control. Our participants were already part of the dance groups. A comparison to no-activity or another group activity would have increased complexity in data interpretation. Based on these issues, we decided against a control group, but it is evident that further studies are needed to understand how individual factors contribute to the observed effects. For instance, in the experimental part, we only used one pop song after the dance classes. One might argue that the lack of an improvement after class with music was due to the choice of the song. It is important to point out though that whilst we did notfind the predicted benefits of music after the class, it did have a facilitating effect before the dancing. Whether the song "Dancing Queen" by ABBA was purely a mismatch after dancing but not before, would require further investigation. Our additional analysis including the TUG indicate that it is more likely that the music was internalised by our participants and thus minimising the relative beneficial effect of the music as an external cue. However, our data is inconclusive on whether a threshold was reached or whether the internalised music beat mismatched and thus interfered with our chosen song. Further studies are required to empirically verify how dancing practice in people with Parkinson's relates to participants' individual differences in their temporal synchronisation abilities.

Finally, our methodological approaches were separate entities and combining phenomenological experiences and quantitative data together is limited. It is however noteworthy how not only the importance of music was emphasised in both parts–but also that the element of the community is flagged up in an unpredicted manner: people with Parkinson's who have attended more dance classes show a stronger benefit of the intervention on motor control. One could argue that this is not only related to functions of motor control but also in some ways by feeling part of a community. Further studies are needed that employ a more controlled approach for the levels of experience. Also, it would be of interest to further assess the observation that music had no effect on movement immediately after a dance class. To evaluate the validity and duration of such a short-term effect, a study with a larger more controlled cohort

is desirable. Moreover, different types of music while performing, such as those matching or mismatching with the beat employed in classes, would shed further light on the underlying processes. Finally, further insight into the role of moving to music could be gained through a follow-on study on the impact of passive music listening.

## Supporting information

**S1 Fig. Influential observations by cook's distance.** Outliers identified by Cook's Distance. Grey line represents the cut-off point at 0.358, corresponding to 4 times the mean. Dotted lines at 1 and 0.5 Coo's $D_i$. X-axis = Observations, y-axis = Cook's distance of the null-model's residuals.
(TIF)

**S2 Fig. Residual plot.** Residuals of null-model before outlier correction.
(TIF)

**S3 Fig. Residual Plot outlier corrected.** Residuals of null-model after outlier correction.
(TIF)

**S1 Table. Descriptive statistics.** Descriptive data reported for Mean Timed Up and Go (TUG) test in seconds and SD for standard deviation for each condition based on the original data without the outlier trials.
(DOCX)

**S1 File. Exploratory analysis with hours of dance class experience as an additional fixed effect.**
(DOCX)

## Acknowledgments

We thank all participants for their invaluable input in sharing their thoughts and experiences for this study. We thank Gemma Coldicott from South London Inclusive Dance Experience (SLiDE), Donna Schoenherr and Beatrice Ghezzi from Move into Wellbeing, Yolanda Aguilar from ShaperCaper as well as Ubit Iskandar and Dido Mirck to provide access to their dance classes for people with Parkinson's. We thank Margriet de Jong for translating the documents into Dutch and our statistician Ahmed Abdullah for statistical advice. We also thank Isaac Duncan-Cross for contextual suggestions.

## Author Contributions

**Conceptualization:** Corinne Jola, Moa Sundström.

**Data curation:** Corinne Jola, Moa Sundström.

**Formal analysis:** Corinne Jola, Julia McLeod.

**Funding acquisition:** Corinne Jola, Moa Sundström.

**Investigation:** Corinne Jola.

**Methodology:** Corinne Jola, Julia McLeod.

**Project administration:** Corinne Jola.

**Resources:** Corinne Jola.

**Supervision:** Corinne Jola.

**Validation:** Corinne Jola, Julia McLeod.

**Visualization:** Corinne Jola.

**Writing – original draft:** Moa Sundström.

**Writing – review & editing:** Corinne Jola, Julia McLeod.

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
