## [Decision Letter · Decision Letter 0]

15 Jun 2022

PONE-D-22-07052Benefits of dance for Parkinson's: The music, the moves, and the companyPLOS ONE

Dear Dr. Jola,

Thank you for submitting your manuscript to PLOS ONE. After careful consideration, we feel that it has merit but does not fully meet PLOS ONE’s publication criteria as it currently stands. Therefore, we invite you to submit a revised version of the manuscript that addresses the points raised during the review process.

We look forward to receiving your revised manuscript.

Kind regards,

J. Lucas McKay, Ph.D., M.S.C.R.

Academic Editor

PLOS ONE

Journal Requirements:

Reviewers' comments:

Reviewer's Responses to Questions

**Comments to the Author**

1. Is the manuscript technically sound, and do the data support the conclusions?

Reviewer #1: Partly

2. Has the statistical analysis been performed appropriately and rigorously? 

Reviewer #1: Yes

3. Have the authors made all data underlying the findings in their manuscript fully available?

Reviewer #1: Yes

4. Is the manuscript presented in an intelligible fashion and written in standard English?

Reviewer #1: Yes

5. Review Comments to the Author

Reviewer #1: This manuscript reads well and has merits. I am concerned about the small sample size, the lack of a control group, the lack of a true pre-post design, and that the mixed methods approach while appreciated may not completely work here. There were two research questions apparently- one about Music and its effects on performance of the TUG. Another was about the felt experience of dancing. I wasn't completely sure how the two matched up. I consider the effect of music to be a research question that can stand on its own. While the authors ran very complex statistics, given the sample size, I wasn't sure this was necessary. The statistical tables are quite thorough but almost provide too much information given what was actually measured.

I wonder if it would be best to split up the paper into two short papers.- One quantitaive and the other qualitative. There's no table 1 with demographics/clinical characteristics. I didn't learn much about who these people with PD were- their stage of PD, their mobility levels, etc.

Minor- in the introduction- action observation therapy is discussed and I really couldn't figure out why. I think the emphasis is on music here, yet so many other threads are brought up that it distracts from the main research questions. I think also that the authors' conclusion that music had no effect post dance class - while interesting- would need to be demonstrated with a much larger better controlled study to reach a definitive conclusion.

It appears also that this effect is what we would call 'short term effect"- given that the effect of no music effect was noted after presumably one dance class- a single dance class was evaluated (I think). Short term or immediate effects have their uses; although I would think a similar experience would need to be done over time, in repeated bouts, to see if this "no music effect" holds.

IN discussion- please do not repeat results, nor provide/reiterate additional statistics. Be very careful about overstatement of results.

A point to consider- Dancing queen has a specific tempo that is pretty honed- there isn't a lot of variability- that would have perhaps have encouraged entrainment to that specific beat. Why would the scores get faster or change at all- if they are walking to the beat? The instructions in TUG here were to walk at the preferred speed not at the 'fast as possible speed'_ which might have elicited some difference.

6. PLOS authors have the option to publish the peer review history of their article (what does this mean?). If published, this will include your full peer review and any attached files.

Reviewer #1: No

---

## [Author Response · Author response to Decision Letter 0]

2 Aug 2022

Reviewer #1 has expressed concerns about the small sample size, the lack of a control group, the lack of a true pre-post design, and the mixed methods approach. We have addressed these points individually below: 

We have sampled according to a cost-effective sample size approach and tested that all assumptions for the linear mixed model were met. 

This means that at the start of the study, we set our sample size to that based on the existing literature. In fact, we aimed to include a larger sample size than comparable studies that measured the effect of a behavioural intervention on the TUG to date (for a recent overview, see Bek et al., 2020). We also calculated the required participant number for a repeated measures design with a small to medium effect size (0.3), a standard power of 0.8 and an alpha err probability of 0.05 using G*power (Faul et al. 2007) which is 24. It is important here to point out that our study followed an experimental behavioural-based approach; it is not a clinical study that compares an intervention group with a control cohort. This is an important difference since a repeated mixed effect design such as ours has greater statistical power than more common between-subjects designs (see also Bek et al., 2020). This means that fewer participants are needed to observe statistically significant effects.

Yet as our data was unbalanced, we decided to run a linear mixed model using LMER which is more appropriate than a repeated measures analysis under this condition. G*power does not allow calculation for linear mixed models, but to address the reviewer’s comment, we run the mixedpower function for mixed linear models in R as illustrated by Kumle and Draschkow (2021). The simulation on the observed data validated that a sample size between 20 and 30 is the optimal participant cohort size as power does not increase with increasing sample size for the fixed effects or the interaction. The power calculation for the interaction of music and time was highest with 0.8 for 26 participants. The only difference to Kumle and Draschkow’s approach is that we lowered the critical t-value to 1 as the threshold. We feel that with a behavioural intervention for which we would not expect any detrimental effect, it is more important to detect an effect for PWP than to miss one. A critical value of 2 is understood to represent alpha of 5% which lowers power overall, but the pattern remains identical.

In the participant section of the manuscript, we have added: “We sampled according to a cost-effective sample size approach with a cohort slightly larger than what can be found in the existing literature with similar designs (see Bek et al., 2020). We also ensured that all the assumptions were met for our analyses.” 

In the discussion section we added: “…since we sampled according to a cost-effective sample size approach, we kept the sample size for the quantitative part of the study small. The possibility of falsely rejecting H1 is thus relatively high. The results of our analyses should therefore be interpreted with caution, and this is why we conducted post-hoc contrast analyses despite the interaction of music and time not reaching significance in the mixed linear effects model.” 

Bek J et al. (2020). Dance and Parkinson's: A review and exploration of the role of cognitive representations of action. Neurosci Biobehav Rev; 109:16-28. doi: 10.1016/j.neubiorev.2019.12.023.

Kumle L, & Draschkow D. (2021). Estimating power in (generalized) linear mixed models: An open introduction and tutorial in R. Behav Res Methods; 53(6):2528-2543. doi: 10.3758/s13428-021-01546-0. 

Faul, F., Erdfelder, E., Lang, A.-G., & Buchner, A. (2007). G*Power 3: A flexible statistical power analysis program for the social, behavioral, and biomedical sciences. Behavior Research Methods, 39(2), 175–191. https://doi.org/10.3758/BF03193146

Regarding the reviewer’s concern of the lack of a control group as well as the lack of a true pre-post design: 

Our participants were assigned to one of four counterbalanced order conditions in a pseudo-randomised order, balancing the number of participants in each group. The randomisation of participants is the most important aspect of a pre-post-test design which we followed accordingly. Further, whilst it is advisable to include a control group in standard pre-post-test designs, it is recognised that this is not without its problems as it increases internal validity - external validity is reduced. Also, selection biases are often an issue. We have considered both problems in the design of our study and decided against a control group. With ecological validity being a core aim of our study, we felt that a control group would not have increased its validity. All our participants were all part of a cohort that have been taking dance classes for some time. This cohort is thus not comparable to a control that is not allowed to partake or is part of another social-activity group. For instance, testing a control cohort that engages in a different group activity would have made the interpretations more complex, also since we yet know so little about the effects of music as an external cue in response to different alternative therapies. Moreover, it is important to emphasise that our study did not focus on evaluating the effectiveness of an intervention per se. Instead, our study investigated the underlying mechanisms of music as an external cue by studying the specific impact of moving with music for a prolonged time has on music as an external cue for movement and whether this aligns with participants subjective experience. Therefore, our study is not a standard intervention study but follows an experimental within-subjects design. Having said that, it would be beneficial to study the impact listing passively to music has for a prolonged time (without moving) on the use of music as an external cue for movement although previous research did not find that participants experienced benefits (Numbela et al., 2013). We have included this in the discussion section: 

“…one could argue that our design misses a control group. Whilst we agree that a control group increases internal validity, external validity, a core focus of our study, is reduced. Another issue is the selection of the control. Our participants were already part of the dance groups. A comparison to no-activity or another group activity would have increased complexity in data interpretation. Based on these issues, we decided against a control group, but it is evident that further studies are needed to understand how individual factors contribute to the observed effects.”

Then, the reviewer felt that a mixed methods approach “may not completely work here” because they understood the two sections as two separate research questions - one about Music and its effects on performance of the TUG and another about the felt experience of dancing. 

We acknowledge that transdisciplinary studies are still uncommon. Yet we argue that qualitative data can support specific interpretations of quantitative findings. In the present study, participants emphasised the relevance of music and most notably, the experience of internalising the music was reported. This corroborates with our quantitative observations and interpretations of the data. It is therefore important to enrich and substantiate quantitative observations with qualitative data particularly for studies on Parkinson’s, with its heterogenous symptomatic. We have made small changes to the manuscript that should clarify the motivation to our approach, and in particular: 

“To advance our understanding of the impact dance classes have on people with Parkinson’s, we thus conducted a mixed-method study that made use of both qualitative and quantitative data to ensure that we capture both quantified behavioural responses and the experience of dance. This is motivated by the recognition that the experience of a dance class is likely to have behavioural implications”

The reviewer commented on the complexity of the statistics used, which we have addressed above in that based on the unbalanced data, this is the appropriate approach. 

The reviewer also highlighted that the statistical tables provided too much information given what was measured. We are grateful for pointing this out and have rectified the information given. At the same time, we also found a minor error in the reporting of the values in the text which we have rectified.

The reviewer expressed missing demographic details/clinical characteristics of the participant cohort. We have originally not included details other than the age and gender distribution since we used the TUG performance as a PD-criteria. This is a linear measurement, and we feel it is therefore a better indicator of the motor stage of PD as required in this study. However, the Hoehn and Yahr scale (HY) is widely accepted and utilised and we have taken note of our participants’ performance according to the HY scale during our study. We have now included this information in the manuscript and report it with further demographic details of our participant cohort for both studies independently. 

The reviewer questioned whether the discussion on the action observation therapy was relevant. We have removed that section and mentioned action observation as an example further down. 

We have also clarified in the conclusion that it would be beneficial to evaluate the no-effect of music post dance class with a larger, better controlled cohort as follows: 

“… it would be of interest to further assess the observation that music had no effect on movement immediately after a dance class. To evaluate the validity and duration of such a short-term effect, a study with a larger more controlled cohort is desirable.”

The reviewer was not happy with our additional statistics in the results section. Exploratory analyses based on the interpretation of the findings (rather than the predictions) are often placed in the discussion section. However, to simplify the narrative, we have moved the particulars to supplementary information and refer to those details in the text instead.

Finally, the reviewer found that our soundtrack “Dancing Queen” might not encourage entrainment due to the lack of variability in rhythm. Our observations at testing were the opposite. This song was liked due to its beat, up-beat melody, and simplicity and participants lightened up when they heard the track, showing signs of entrainment before testing. We argue that the reason as to why participants walk faster or slower is because firstly, not everyone has perfect ability to recognise a beat and secondly, walking to the beat is not necessarily achievable even if they try to. We emphasised in our instructions that the speed is important and that we do measure time. We instructed participants to walk at the preferred speed to gather their natural response (where some can keep up with the beat with most steps whilst others cannot) without increasing a risk of falls. We have clarified this in the manuscript.

---

## [Editor Report · Decision Letter 1]

19 Oct 2022

PONE-D-22-07052R1Benefits of dance for Parkinson's: The music, the moves, and the companyPLOS ONE

Dear Dr. Jola,

Thank you for submitting your manuscript to PLOS ONE. After careful consideration, we feel that it has merit but does not fully meet PLOS ONE’s publication criteria as it currently stands. Therefore, we invite you to submit a revised version of the manuscript that addresses the points raised during the review process. Some minor concerns remain that the journal would like to have addressed prior to publication.

1. The notion of a "cost-effective" sample size is not really supported here. Like most early studies, it appears that the sample size was limited by resource limitations and other practical factors, and therefore no a priori power analysis was performed. Please remove this language and replace it with something like "Due to resource limitations it was not practical to set the sample size based on an a priori power analysis. Our sample was therefore a convenience sample." Also please reference in the limitations that post hoc power analyses suggested that the study was underpowered, and therefore these results would benefit from replication.

2. It appears that the linear model without interaction yielded two statistically-significant effects but that adding the interaction term caused the p values to increase beyond the cutoff value. The authors then justify using one or the other of the models with an information criterion argument. This is fine. However, it is unclear to which model "full model" refers. Please edit the captions of tables 3 and 4 to describe the model used, e.g., "these values reflect a linear mixed model with fixed effects for Music and Time."

3. Please provide a table of the original outcome measure values (e.g., TUG times, in seconds, mean ± SD) in the supplement. No statistical tests are required.

4. The regression diagnostics plots are not necessary for this work and are distracting. Please move them to the supplement. Please ensure that your decision is justified on PLOS ONE’s publication criteria and not, for example, on novelty or perceived impact.

We look forward to receiving your revised manuscript.

Kind regards,

J. Lucas McKay, Ph.D., M.S.C.R.

Academic Editor

PLOS ONE
---

## [Author Response · Author response to Decision Letter 1]

1 Nov 2022

Response to reviewers

We were asked to #1 clarify the notion of our sample size, to #2 clarify the model and table values, to #3 provide a table with mean ±SD, and to #4 move the diagnostics plot into the supplement. We have addressed all points in the manuscript and clarify our changes below: 

#1 We have now clarified the resource limitations by including the following two sentences in the participant section: “Due to resource limitations, it was not practical to set a sample size based on a priori power analysis. Our sample was therefore a convenience sample.” 

And in the limitations, we rephrased our section on the participant sample size. Please note that we feel that including the phrase that “the study is underpowered” does not provide an accurate description of what power is or does. It now reads: “Another limitation of our study is that that we had resource limitations that restricted the number of participants. It can thus be argued that our design is underpowered – or in other words, less sensitive to our alternative hypotheses H1, H2, and H3. The risk for falsely rejecting small to medium effects is indeed relatively high with a small sample size. The risk for falsely rejecting small to medium effects is indeed relatively high with a small sample size. The results of our analyses should thus be interpreted with caution when even small effects are effects of interest as they are here;…… Clearly, our results would benefit from replication in form of a clinical trial.

To clarify our stance: we agree that in psychological research, power analyses are an important tool. It allows to determine the smallest sample size that is suitable to detect an effect of a particular size for a given test at the desired level of significance. Notably, our G*power analysis (described in our previous response) showed that for a small to medium effect, our study was not underpowered. In general, there is however an argument that post-hoc power analyses should not be conducted. Also note that dealing with power in linear mixed model analyses is different altogether. Firstly, it is not common to conduct a power analysis for mixed linear model analyses. They are so powerful because they deal better with variation in the data than standard repeated measures designs. Secondly, we have previously explained the specific issue of our relatively small sample size in that it increases the risk for falsely rejecting our H1. 

We have now added information and changed this part (see above highlighted section) regarding issues of the small sample size and what that means for the interpretation of our data. We feel that our response provides the reader with more specific information regarding the challenges of a relatively small sample size which we feel addresses the PLOS One’s publication criteria better than a more general description of a study being underpowered. 

#2 There seems to have been some confusion regarding the terminology of the “null-model” and the “alternative model”. We thank the reviewer for pointing this out and have now changed this. It reads: “When normality of the underlying residuals could be assumed, we compared the null-model with the alternative ‘full-model’, including the predictors conditions music and time as fixed effects and participants as a random nested effect (M*T + (1|p)).” We have also clarified all references to the two models throughout the manuscript. 

As for the tables, please note that table 3 is the descriptive statistics that we were asked to include. The values describe the data and are not representing a model and table 4 shows contrasts and pairwise effects according to least-squares means which are also not representing a model. Therefore, it would be incorrect to state that “these values reflect a linear mixed model….” but because other readers may have the same uncertainty about the tables, we have added further information regarding for which model the values were obtained.

#3 We are not sure why the reviewer asks us for this additional data since all data from our analyses are available in sufficient detail in the manuscript and the tables as well as the raw data which is accessible through the OSF (Open Science Framework). We think it is important to highlight that mean and SD are not original outcome measure values of a linear mixed model analysis. Nevertheless, since other readers may have similar doubts, we have created a table with what we think the reviewer asked us for and included this in the supplement (S2).

#4 We agree that the regression diagnostics plots are better placed in the supplement and have moved them there.

---

## [Editor Report · Decision Letter 2]

4 Nov 2022

Benefits of dance for Parkinson's: The music, the moves, and the company

PONE-D-22-07052R2

Dear Dr. Jola,

We’re pleased to inform you that your manuscript has been judged scientifically suitable for publication and will be formally accepted for publication once it meets all outstanding technical requirements.

Kind regards,

J. Lucas McKay, Ph.D., M.S.C.R.

Academic Editor

PLOS ONE